# WISP2/CCN5 Suppresses Vasculogenic Mimicry through Inhibition of YAP/TAZ Signaling in Breast Cancer Cells

**DOI:** 10.3390/cancers14061487

**Published:** 2022-03-14

**Authors:** Nathalie Ferrand, Aude Fert, Romain Morichon, Nina Radosevic-Robin, Maurice Zaoui, Michèle Sabbah

**Affiliations:** 1Team Cancer Biology and Therapeutics, Centre de Recherche Saint-Antoine (CRSA), Institut Universitaire de Cancérologie, Sorbonne University, Inserm UMR_S 938, 75012 Paris, France; nathalie.ferrand@inserm.fr (N.F.); aude.fert@sorbonne-universite.fr (A.F.); maurice.zaoui@inserm.fr (M.Z.); 2CRSA Cytométrie Imagerie Saint-Antoine, Sorbonne University, 75012 Paris, France; romain.morichon@sorbonne-unversite.fr; 3Centre Jean Perrin, Department of Pathology, University Clermont Auvergne, INSERM U1240 (IMoST), 63011 Clermont-Ferrand, France; nina.robin@clermont.unicancer.fr; 4Centre National de la Recherche Scientifique (CNRS), 75016 Paris, France

**Keywords:** WISP2/CCN5, vasculogenic mimicry, breast cancer, CYR61, YAP-TAZ signaling

## Abstract

**Simple Summary:**

Breast cancer is the most frequent malignancy in women worldwide. Advanced breast cancer with distant organ metastases is considered incurable with currently available therapies. The vasculogenic mimicry (VM) process is associated with an invasive and metastatic cancer phenotype and a poor prognosis for human breast cancer patients. Our aim was to study the effect of WISP2, a matricellular protein, on VM. We found that WISP2 inhibits VM through inhibition of CYR61 protein expression and YAP-TAZ signaling. Our finding may open promising candidates for blocking VM in breast cancer.

**Abstract:**

Vasculogenic mimicry (VM) formed by aggressive tumor cells to create vascular networks connected with the endothelial cells, plays an important role in breast cancer progression. WISP2 has been considered as a tumor suppressor protein; however, the relationship between WISP2 and VM formation remains unclear. We used the in vitro tube formation assay and in vivo immunohistochemical analysis in a mouse model, and human breast tumors were used to evaluate the effect of WISP2 on VM formation. Here we report that WISP2 acts as a potent inhibitor of VM formation in breast cancer. Enforced expression of WISP2 decreased network formation while knockdown of WISP2 increased VM. Mechanistically, WISP2 increased retention of oncogenic activators YAP/TAZ in cytoplasm, leading to decreased expression of the angiogenic factor CYR61. Studies using an in vivo mouse model and human breast tumors confirmed the in vitro cell lines data. In conclusion, our results indicate that WISP2 may play a critical role in VM and highlight the critical role of WISP2 as a tumor suppressor.

## 1. Introduction

Breast cancer is the leading cause of cancer death in women due mainly to its ability to metastasize to vital tissues. The development and progression of breast cancer is a complex process involving hormonal factors, genetic and epigenetic alterations.

It is widely accepted that tumors require blood supply to survive, grow, and metastasize [1]. This concept has been inextricably linked to angiogenesis, a process that corresponds to the growth of new blood vessels within a tumor. Although, angiogenesis is an important mechanism for tumor growth, survival, and metastatic processes, anti-angiogenic drugs were not effective in all cancers, and resistance to these drugs could occur [2,3]. However, it is now well established that tumor vasculogenesis is not necessarily attributed to endothelial cells alone. Indeed, in some tumors, cancerous tissues may become vascularized by the networks created by the tumor cells themselves through their acquisition of plasticity to mimic endothelial function, a phenomenon called vascular mimicry (VM) [4]. VM was first reported in melanoma and is an alternative way to provide sufficient blood perfusion for highly invasive and metastatic phenotypes in many cancers [5,6]. The network of VM is rich in extracellular matrix and independent of endothelial cells. A vascular mimicry–angiogenesis junction has been suggested based on the presence of blood flow in the vascular channels [7].

WNT1 inducible signaling pathway protein-2 (WISP2), also known as CCN5 or COP1, belongs to the connective tissue growth factor/Cysteine-rich 61/Nephroblastoma overexpressed (CCN) protein family, a family of six secreted proteins involved in many physiological and pathophysiological processes, such as development, cell proliferation, angiogenesis, and tumorigenesis [8,9]. 

Mounting studies have identified that WISP2 is critically involved in tumor cell invasion and metastasis in breast cancer [10]. Loss of WISP2 expression is associated with epithelial-to-mesenchymal transition (EMT) and the emergence of a cancer stem-like cell phenotype [11]. Interestingly, the transcriptional signature of VM shares components with that of stemness and EMT, key attributes involving tumor plasticity during metastasis and resistance to chemotherapy [12,13]. WISP2 has a special position within the CCN family; it is the only member that lacks the CT module and has been described as playing a negative dominant role vis-à-vis other family members [14]. Indeed, while WISP2 is described as a negative regulator of migration and invasion of mammary tumor cells, four members of this family, CYR61, CTGF, NOV, and WISP1, are described as positive regulators overexpressed in invasive and metastatic cancers [15,16,17]. Moreover, they exhibit pro-angiogenic activities and are important regulators of endothelial cell function [18,19]. Furthermore, in cancer models CYR61 and CTGF are involved in the formation of new blood vessels via vasculogenic mimicry [20,21]. In contrast, WISP2 was found to rather inhibit the proliferation of vascular smooth muscle cells suggesting an opposite relation to vascular biology [22]. 

Moreover, based on microarray analysis on WISP2-negative breast cancer cells, one of the pathways that we found specifically activated is the angiogenesis pathway [23]. Recent studies have shown that cancer stem cells are capable of trans-differentiating into endothelial cells both in vivo and in vitro and thus contribute to VM [24,25].

These studies prompted us to investigate whether WISP2 may also play a key role in influencing VM. The aims of our study were to compare the ability of human breast cancer cells expressing or not expressing WISP2 to form VM on three-dimensional Matrigel cultures in vitro and by immunohistochemical analysis in mouse models and human breast tumors in vivo. In additional, we sought to identify candidate and molecular mechanisms that are involved in channel formation. 

## 2. Materials and Methods

### 2.1. Cell Lines

Human breast carcinoma MCF7 and MDA-MB-231 cell lines were purchased from the ATCC (American Type Culture Collections, VA, USA). MCF7-sh-WISP2 cell line was established by transfection of MCF7 cells with WISP2-directed sh-RNA plasmid as previously described [10], and this cell line was named sh-WISP2. MDA-MB-231 cells were transfected with pCEP4-Flag vector or pCEP4-Flag-WISP2 vector expressing full-length human WISP2 as previously described [26]. The stable transfectants established were named w6 and w15. All these cells were maintained in DMEM (Dulbecco modified Eagle’s medium) supplemented with 10% (*v*/*v*) FBS (fetal bovine serum). Human dermal Microvascular Endothelial Cells, HMEC-1 were cultured in MCDB 131 medium supplemented with 10% (*v*/*v*) FBS, L-Glutamine 1%, 10 ng/mL human recombinant EGF, and 1 µg/mL hydrocortisone.

### 2.2. Real-Time RT-PCR

Total RNA was extracted from all cell lines using the TRIzol^®^ RNA purification reagent. RNA quantity and purity were determined by using a Spectrophotometer DS-11 (Denovix, Wilmington, DE, USA). One microgram of total RNA from each sample was reverse transcribed, and real-time RT-PCR measurements were performed as described previously [11] using an apparatus Aria MX (Agilent Technologies, Santa Clara, CA, USA) with the corresponding SYBR^®^ Green kit, according to the PROMEGA manufacturer’s recommendations. The mRNA levels indicated below show the abundance of the target gene relative to that of two endogenous controls (β-Actin and RPLP0, also known as 36B4) used to normalize the starting amount and quality of total RNA. 

### 2.3. Western Blot and ELISA

Cell extracts were obtained after lysis with RIPA buffer (ref 89901, Pierce) supplemented with protease (B14001) and a phosphatase inhibitor cocktail (B15001, Bimake, Euromedex); and equal amounts of protein were loaded onto SDS-PAGE gels. After transfer onto nitrocellulose membrane, blots were incubated overnight at 4 °C with the appropriate antibody, followed by incubation with a horseradish peroxidase-conjugated secondary antibody (1/2000, Cell Signaling, Danvers, MA, USA). Bands were visualized using the Clarity™ Western ECL substrate on Chemidoc systems (Bio-Rad, Hercules, CA, USA) [27]. Immunoblot analyses were performed by using antibodies directed against WISP2 (ab38317, Abcam); antibodies directed against CYR61 (26689-1-AP), MMP14/MT1-MMP (14552-1-AP), MMP9 (10375-2-AP), Endoglin/CD105 (10862-1-AP), YAP (13584-1-AP), TAZ (23306-1-AP), and TEAD1 (13283-1-AP) were purchased from Proteintech; EphA2/D4A2 (#6997), VEGFR2 (#2479), and β-Actin/13E5 HRP (#5125) were obtained from Cell Signaling, and FLAG^®^ M2 (F1804) from Sigma. Protein expression was quantified by densitometric analysis of the immunoblots using Image Lab software developed by Bio-Rad. CYR61 concentrations in the conditioned media were determined by using Human CYR61/CCN1 Quantikine Elisa according to the manufacturer’s instructions (DCYR10; Bio-Techne, San Jose, CA, USA).

### 2.4. Matrigel Tube Formation

Tube Formation Assays were performed using µ-Slide Angiogenesis (Ibidi 81506, Biovalley, Illkirch-Graffenstaden, France) pre-coated with 10 µL of Matrigel^®^ Growth Factor Reduced, Phenol Red-Free (#356231, Corning, New York, NY, USA) and were allowed to polymerize at 37 °C for at least 30 min. Cells (2 to 3 × 10^5^ cells/mL) were seeded into each well in triplicate and maintained in appropriate medium supplemented with 2% (*v*/*v*) FBS. A time-lapse video was recorded for 24 h to follow the formation of the tubules, using an IX83 inverted microscope (Olympus, Tokyo, Japan) with a motorized stage and an equipped CO_2_ thermostat chamber (Pecon cell vivo). The images were based on tube length and analyzed with Angiogenesis Analyzer for ImageJ [28].

### 2.5. Cell Viability Assay and Verteporfin Treatment

MDA-MB-231 cells were added into 12-well plates for cell viability measured by MTT as previously described [29]. Cells were treated with 0, 4, or 7 µM Verteporfin (SML0534, Sigma-Aldrich, Saint-Louis, MO, USA) over 24 h.

### 2.6. Immunofluorescence Staining

Cells were plated on chamber slides and fixed in 4% paraformaldehyde. Cells were stained with anti-YAP antibody (1:50) and secondary anti-Rabbit CyTM3-conjugated antibody (1:200, 711-165-152, Jackson ImmunoResearch, Cambridgeshire, UK). After immunolabeling, cells were washed, stained with 1 µg/mL DAPI (Sigma), and viewed by fluorescence microscopy (BX61, Olympus).

### 2.7. Immunohistochemistry

Immunochemical staining was performed on 5 µm-thick sections of formalin-fixed, paraffin embedded (FFPE) orthotopic xenograft mouse tumors obtained after inoculation with human MCF7-sh-WISP2 or MDA-MB-231 cells [22]. Sections were deparaffinized in xylene and hydrated in graded ethanol, then treated with Signal Stain EDTA Unmasking Solution (#14747, Cell Signaling Technology, Danvers, MA, USA) at 100 °C for 30 min. Subsequently, the slides were incubated with anti-CD31 (1/50; ab28364, Abcam, Cambridge, MA, USA) or anti-CD31 (1/3000; 11265-1-AP, Proteintech, Rosemont, IL, USA), anti-CYR61 (1/200; 26689-1-AP, Proteintech, Rosemont, IL, USA) or anti-YAP (1/100; 13584-1-AP, Proteintech, Rosemont, IL, USA). The antigen-antibody reaction was visualized by NeoStain ABC Kit HRP (NB-23-00001-6, NeoBiotech, Rosemont, IL, USA). The slides were rinsed with water for 1 min to stop the DAB staining reaction. Finally, the slides were treated with Periodic Acid Solution (3951, Sigma-Aldrich, Saint-Louis, MO, USA ) for 5 min and rinsed with distilled water for 5 min. In a dark chamber, the slides were treated with Schiff’s reagent (3952016, Sigma-Aldrich, Saint-Louis, MO, USA ) for 15 min, rinsed with distilled water and counterstained with Hematoxylin (GHS316, Sigma-Aldrich, Saint-Louis, MO, USA ). The slides were mounted with glycerol gelatin (GG-1, Sigma-Aldrich, Saint-Louis, MO, USA).

Eleven representative primary triple negative breast cancer FFPE samples were obtained from the Department of Pathology, Centre Jean Perrin in Clermont-Ferrand (Clermont-Ferrand, France). Four µm sections were used for double IHC labeling (CYR61/PAS or CD31/PAS) using anti-CD31 (1/3000; 11265-1-AP, Proteintech, Rosemont, IL, USA) and anti-CYR61 (1/200; 26689-1-AP, Proteintech, Rosemont, IL, USA) and the same protocol as the mouse tissue sections. 

All the mouse and human stained sections were scanned and analyzed with the CaseViewer digital microscopy application, to evaluate spatial expression of CYR61, CD31, and YAP distribution. 

Human tumor cell expression of CYR61, CD31, and YAP was assessed semi-quantitatively by measuring the area of labeled cells using the ImageJ program and scored as follows: (a) 1 = <5%; (b) 2 = 5–49%; (c) 3 = >50% of tumor surface occupied by the stained tumor cells. 

### 2.8. Statistical Analysis

Data are shown as the average of ± SEM for more than 3 independent experiments. As differences between test and control conditions do not assume Gaussian distribution, test and control conditions were assessed using the nonparametric Mann–Whitney test analysis. Significance is indicated by: * when *p* < 0.05, ** when *p* < 0.01 and *** when *p* < 0.005.

## 3. Results

### 3.1. WISP2 Reduces Angiogenic-Associated Gene Expression

To further explore the mechanism involved in the WISP2-mediated anti-angiogenic effect, several important angiogenic-associated genes identified on microarray analysis [23] in sh-WISP2 cells were evaluated by quantitative RT-PCR and Western blotting (Figure 1A,B). We then compared the mRNA and protein expression levels of selected genes known to be involved in VM [30,31] in the MDA-MB-231 cell line that do not express WISP2 (Figure 1A and inset) and exhibiting mesenchymal and stemness phenotype. As shown in Figure 1A,B, we observed a high expression of these six genes in sh-WISP2 and MDA-MB-231 cells compared to MCF7 cells at mRNA and protein levels.

### 3.2. WISP2 Suppresses Vasculogenic Mimicry

To investigate whether WISP2 affects the formation of vessels-like networks, we utilized an in vitro VM assay that assesses cell network formation on Matrigel [32]. Three-dimensional cultures were conducted to estimate the ability of the vessel-like channels formation in MCF7, sh-WISP2, and MDA-MB-231 breast cancer cells using HMEC-1 endothelial cells as positive control. Time-lapse video microscopy experiments revealed that sh-WISP2 and MDA-MB-231 cells showed ability to form channels in a similar manner to HMEC-1 cells (Figure 2A and Appendix A). In contrast, no channels appeared in MCF7 cells even when the incubation time was prolonged to 24 h (Figure 2A). To determine whether the VM observed in vitro could be observed in vivo, sh-WISP2 and MDA-MB-231 cells were injected into the mammary fat pad of nude mice. Analysis of sections of tumors derived from sh-WISP2 and MDA-MB-231 cells revealed a network of VM identified by CD31 negative and PAS positive double staining (Figure 2B)

To further assess the role of WISP2 in this process, we added recombinant human WISP2 to endothelial or breast tumor cells lacking WISP2. When sh-WISP2, MDA-MB-231, and HMEC-1 cells were treated for 48 h with recombinant WISP2 before being plated on Matrigel, we found that WISP2 treatment was able to significantly reduce the number of vascular channels formed by these cells as compared with cells treated with vehicle alone (Figure 3A and Appendix A). On the other hand, the MDA-MB-231 derived cell lines w6 and w15 overexpressing WISP2 [20] were tested for their VM ability. Compared to the control cell lines (cells transfected with vehicle vector), w6 and w15 constitutively overexpressing WISP2 tended to form 50% less vascular-like tubes on Matrigel, suggesting a role for WISP2 in channel formation (Figure 3B and Appendix A).

### 3.3. WISP2 Downregulates CYR61 Expression

Because WISP2 may act as a dominant-negative regulator of other CCN family members, we next investigated the impact of WISP2 on the expression levels of CYR61 known as a pro-angiogenic factor [33,34]. We measured 13- to 15-fold higher levels of secreted CYR61 in MDA-MB-231 and sh-WISP2 cells, respectively, as compared to MCF7 cells, and even 6- to 8-fold compared to HMEC-1 cells (Figure 4A). Interestingly, following overexpression of WISP2, levels of CYR61 were decreased by approximately 50 to 70% (Figure 4A). Moreover, the addition of human recombinant WISP2 (Figure 4B) or overexpressed WISP2 (Figure 4C) induces a decrease of the mRNA and protein levels of CYR61 in the MDA-MB-231 cell line. These results suggest that WISP2 functions as a negative regulator of VM, which is associated with downregulation of CYR61 protein. 

### 3.4. WISP2 Regulates the Hippo/YAP Pathway

To unravel the mechanism(s) underlying WISP2-regulated VM, we examined the Hippo/YAP/CYR61 signaling axis that is critical for angiogenesis and VM. As shown in Figure 5A, the levels of total YAP, TAZ, and TEAD were upregulated in MDA-MB-231 and sh-WISP2 compared to MCF7 (Figure 5A). To examine the role of YAP during VM, we used verteporfin, a pharmacological inhibitor of YAP/TAZ. Administration of verteporfin in MDA-MB-231 cells effectively reduced tube formation (Figure 5B). Importantly, the reduced VM caused by YAP/TAZ inhibition is not due to cell death (Appendix A). Furthermore, we showed a 10-fold decrease of CYR61 mRNA and protein levels when MDA-MB-231 cells were treated with verteporfin (Figure 5C). Finally, we examined the subcellular localization of YAP after overexpression of WISP2 in MDA-MB-231 cells. More than 50% of cells showed a nuclear localization of YAP in controls cells (Figure 5D). In addition, overexpression of WISP2 was significantly correlated with a cytoplasmic localization of YAP (70% of cells), suggesting a decrease of the transcriptional activity of YAP (Figure 5). 

To confirm the above in vitro cell line data in an in vivo mouse model, sections of tumor derived from xenografts in nude mice after injection of MDA-MB-231 or sh-WISP2 human breast cell lines were analyzed. CYR61 and YAP labeling associated to PAS staining were performed (Figure 6A). Microscopic observations showed that VM channels positive for Periodic Acid-Schiff are lined by tumor cells, positive for CYR61 or YAP labeling.

To validate this finding in breast cancer patients, we examined VM in tumor specimens surgically removed from triple negative breast cancer. Samples were selected from a previously published cohort of triple negative breast cancers [35]. CYR61, CD31 and YAP expression profiles were assessed by IHC in all specimens (Figure 6B) and revealed a strong CYR61 and YAP protein expression in the cytoplasm of the cells bordering PAS-labeled mimetic vascular channels; on the other hand, the same cells were negative for CD31, demonstrating that they are not blood vessels. On the basis of IHC scoring, CYR61 and YAP were found to be strongly expressed (score = 3); moreover, this result was observed in more than 60% of the patient sections analyzed.

## 4. Discussion

Although several lines of evidence suggest that WISP2 has an anti-invasive role in carcinogenesis [10,14,36], the mechanisms of WISP2 in breast cancer have not been thoroughly investigated. Multi mechanisms have been proposed to be involved in the WISP2-mediated anti-invasive activity in breast cancer. For instance, WISP2 may suppress EMT in breast cancer [10,36,37,38], on the other hand WISP2 silencing promotes a stem-like cell phenotype [11] and the loss of the IGF1 and EGF mitogenic effect in ER-α positive breast cancer cells [39,40]. Furthermore, EMT is an important event involved in both metastasis and a new vascular paradigm called vasculogenic mimicry (VM) corresponding to the capacity of aggressive tumor cells to form vessel-like networks. In the present study, we provide evidence that WISP2 could impair VM formation to inhibit tumor progression via its intracellular action to inhibit Hippo/YAP/TAZ signaling pathways, and subsequently leads to genetic decrease of CYR61.

First, we found that only breast cancer cells that do not express WISP2 form vascular networks in vitro and in vivo. Second, we show that both endogenous WISP2 overexpression and exogenous WISP2 treatment decreased tube formation. To our knowledge, the present study is the first report to demonstrate that WISP2 plays a crucial role in breast VM formation and consequently contributes to the inhibition of tumor progression. 

Studies have revealed a unique role of CCN proteins, particularly CYR61, CTGF, and NOV in modulating developmental, physiological, and pathological angiogenic events [18]. Here, we show that loss of WISP2 expression leads to a dramatic increase in CYR61 expression and that recombinant WISP2 proteins could downregulate the expression of CYR61. CYR61 is a key factor in mesenchymal stem cell secretome that contributes to the angiogenic response [41] and enhances neovascularization and tumor formation of human tumor cells in immunodeficient mice [17,33,42], and it has been proposed that CYR61 is an angiogenic factor. 

YAP/TAZ are established as critical regulators of developmental angiogenesis [43,44]. In the pathological angiogenic model, many genes and signaling pathways have been shown to regulate angiogenesis and vascular mimicry though YAP/TAZ [45,46] and interestingly, CYR61 is a transcriptional target of YAP/TAZ [47]. Mounting evidence has indicated that the Hippo/YAP/TAZ signaling pathway is implicated in breast cancer [48,49]. When the Hippo pathway is active, yes-associated protein 1 (YAP1) and the transcriptional coactivator with PDZ-binding motif (TAZ) are phosphorylated by Hippo core kinases, large tumor suppressor 1 and 2 (LATS1/2), resulting in their cytoplasmic retention and proteasomal degradation. On the contrary, when the pathway is off, YAP1 and TAZ are nuclear and preferentially bind to the TEA domain transcription factor (TEAD) to promote cell survival and proliferation through expression of target genes, including the connective tissue growth factor (CTGF) and cysteine-rich angiogenic inducer 61 (CYR61) [50]. Here, we found that the expression levels of YAP, TAZ, TEAD and their target gene CYR61 are inversely associated with that of WISP2. Furthermore, pharmacological inhibition of YAP/TAZ with verteporfin reduced tube formation. Interestingly, the Hippo pathway inhibits Wnt/β-catenin signaling [51]. In YAP-downregulated colorectal cancer cells, CTGF was downregulated, whereas WISP1, a Wnt-target gene, is upregulated [51]. However, a few regulatory factors and cellular processes acting on the Hippo pathway have been uncovered (actin dynamics, cell matrix stiffness, cell–cell contact, and lysophosphatidic acid) [52]. 

Many studies have revealed some discrepancies in the role of WISP2 in various cancers and could explain the dual role of WISP2 in the Hippo/YAP/TAZ signaling. Moreover, a recent study examining the effect of LATS1/2 deletion in colon cancer cell growth showed an inhibition of cell proliferation in vitro and tumor growth in vivo, together with a selective induction of WISP2 gene expression [53]. Furthermore, WISP2 was identified as a direct target of YAP/TAZ [53]. In contrast, WISP2 deletion inhibited ovarian cancer cell proliferation and activated YAP in vivo and in vitro [54]. The activity of YAP/TAZ is regulated by physical interactions with other proteins which sequester YAP/TAZ in the cytoplasm and inactivate it [55,56]. In the present study, overexpression of WISP2 in cells leads to the accumulation of cytoplasmic YAP leading to inhibition of YAP/TAZ activity and a decrease in CYR61 gene expression. We have confirmed the presence of VM in the aggressive triple negative subtype of breast cancer and revealed a strong expression of CYR61 and YAP in human tumor cells but not in CD31 positive endothelial cells.

Although angiogenesis is crucial for tumor growth, the effect of anti-angiogenic monotherapy appears to be limited in breast cancer [57]. Some studies have shown that resistance to anti-angiogenic therapy could be due to VM formation, so targeting the VM is a therapeutic challenge. There is a lack of effective anti-VM drugs in clinical trials; however, various molecular pathways underlying VM have been proposed, such as VEGF, NF-κB, and PI3K [58]. In the present study, we found that inhibiting YAP/TAZ activity could inhibit VM formation, and our finding that WISP2 could decrease VM formation via inhibition of YAP/TAZ signaling may help to develop a novel therapy. 

## 5. Conclusions

In conclusion, our study demonstrated an important role of WISP2 in vascular mimicry formation in breast cancer. We provided evidence that WISP2 negatively correlates with VM formation in vitro and in vivo. We showed that WISP2 inhibited tube formation by modulating the Hippo/YAP/TAZ signaling pathway resulting in a downregulation of CYR61. Therefore, inhibiting YAP/TAZ- TEAD is an attractive and viable option for novel cancer therapy. It is exciting to know that many drugs already in the clinic restrict YAP/TAZ activities, and several novel YAP/TAZ inhibitors are currently under development. Further investigations to decipher the molecular basis underlying the effects of WISP2 are needed in order to design effective therapeutic options to suppress VM formation in breast cancer.

## Figures and Tables

**Figure 1 cancers-14-01487-f001:**
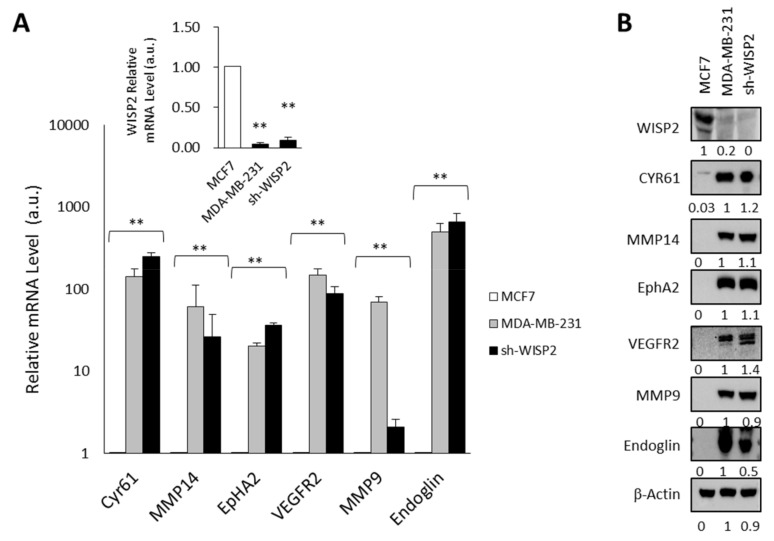
Loss of WISP2 induces angiogenic-associated gene expression (**A**) RNA was isolated from human breast cancer cell lines and angiogenic gene expression was analyzed by qRT-PCR. The results, after normalization as described in Materials and Methods, represent the relative transcript levels among these different cell lines tested and are expressed as mean ± SEM from three independent experiments. The inset shows WISP2 mRNA levels. The values indicate the changes for the indicated samples compared to MCF7 cell line. ** *p* < 0.01 (**B**) Protein extracts of different cell lines were prepared and tested by Western blotting for angiogenic protein expression. The levels of β-actin in cell lysates were measured by Western blotting and included as a loading control. Levels of proteins were calculated by densitometry and listed beneath the bands. The original western blots can be found in Appendix A.

**Figure 2 cancers-14-01487-f002:**
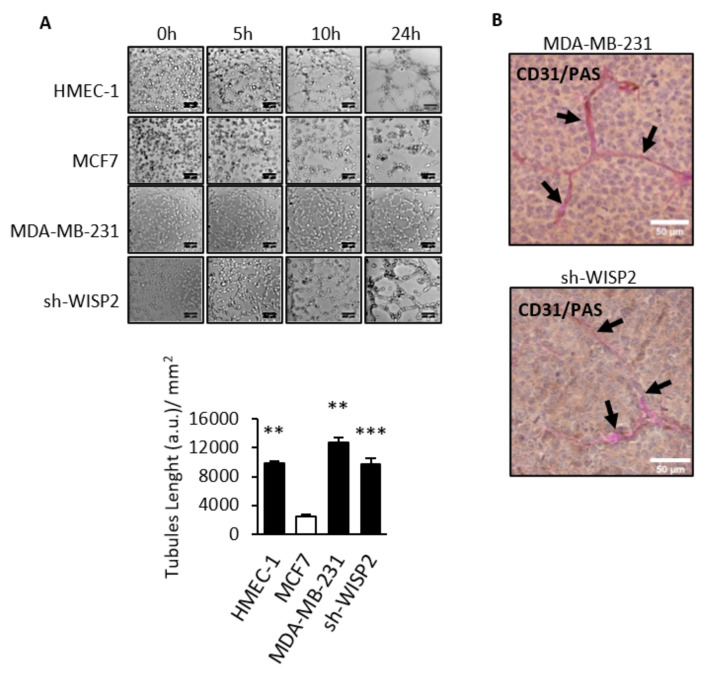
Expression of WISP2 correlates with vasculogenic mimicry (VM). Time-lapse video microscopy experiments were realized on endothelial HMEC-1 cells and breast cancer MCF7, MDA-MB-231, and sh-WISP2 cells. (**A**) Representative images undergoing VM and quantitation of VM length of tubes after 24 h. Experiments were realized at least 3 times and data are expressed as mean ± SEM. Scale bar = 200 μm. (**B**) Thick paraffin sections of breast obtained from tumor xenografts of nude mice injected with MDA-MB-231 or sh-WISP2 human breast cell lines were subjected to CD31-Periodic Acid-Schiff (PAS) dual staining. Black arrows indicate VM channels with positive PAS and negative CD31 expression. (Scale bar = 50 µm). ** *p* < 0.01; *** *p* < 0.005.

**Figure 3 cancers-14-01487-f003:**
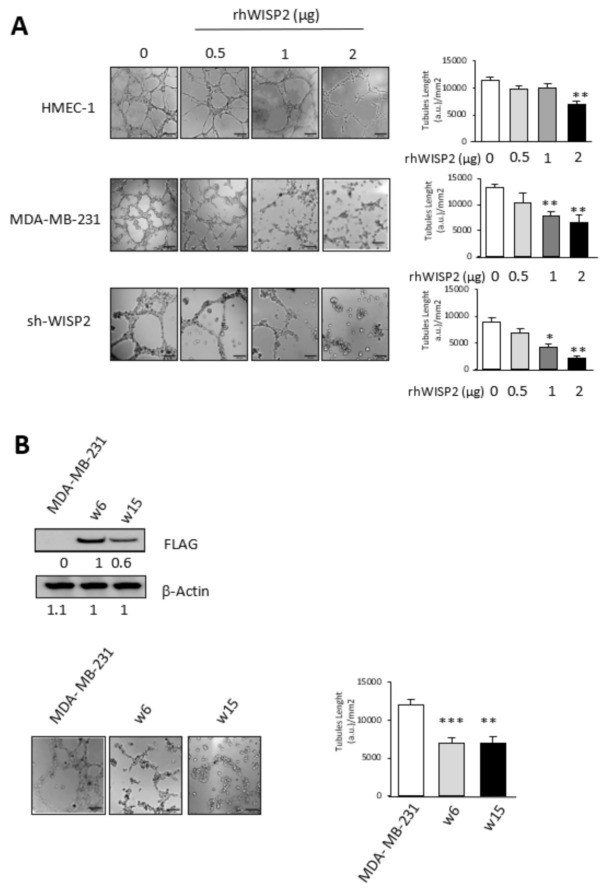
WISP2 inhibits VM tubes formation (**A**) HMEC-1 endothelial cells, and breast cancer MDA-MB- 231 and sh-WISP2 cells were pre-treated for 48 h with 0.5, 1, and 2 µg of human recombinant WISP2 (rhWISP2) before being seeded on Matrigel. Representative images of VM were observed over 24 h, and length of tubes quantified. (**B**) Western Blot showing overexpression of WISP2 in two clones (w6 and w15) of MDA-MB-231 stably transfected with Flag-WISP2 expressing vector. Representative images and quantitation of length of tubes formation in w6 and w15 cell lines. Levels of proteins were calculated by densitometry and listed beneath the bands. Scale bar = 200 μm. * *p* < 0.05; ** *p* < 0.01; *** *p* < 0.005. The original western blots can be found in Appendix A.

**Figure 4 cancers-14-01487-f004:**
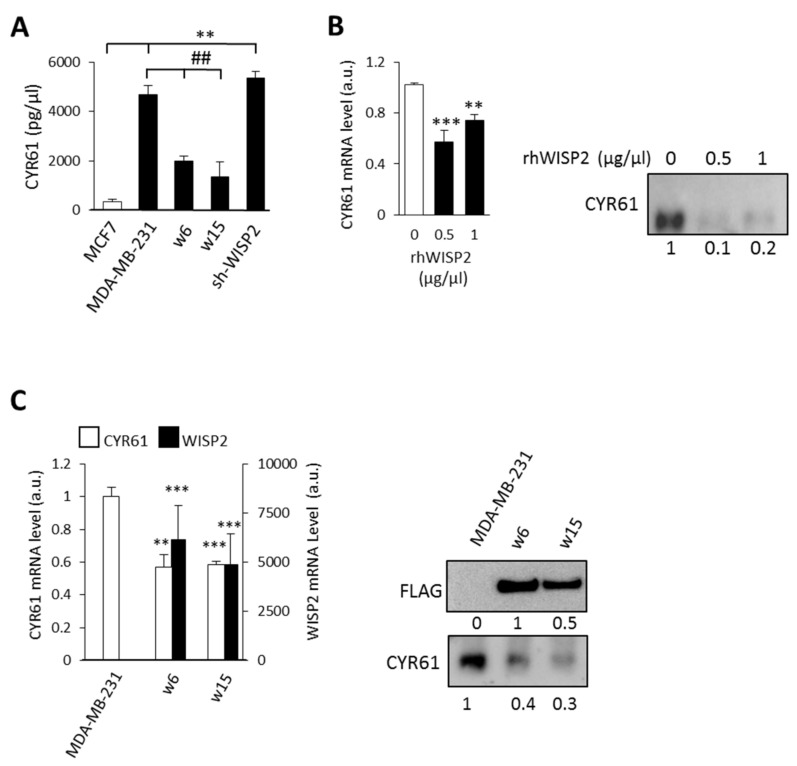
WISP2 downregulates CYR61 expression. (**A**) Conditioned medium from human breast cancer cell lines was analyzed for CYR61 expression by ELISA. CYR61 concentration was normalized by the total protein levels. (**B**) MDA-MB-231 cells were incubated for 48 h in the presence of increasing concentrations of recombinant WISP2. CYR61 mRNA and protein expression were analyzed by qRT-PCR and Western blotting. (**C**) CYR61 and WISP2 mRNA and protein expression were analyzed by qRT-PCR and Western blotting in MDA-MB-231 w6 and w15 cell lines overexpressing WISP2. Immunoblot signals were quantified by densitometry; all values are representative of data from 3 independent experiments. Levels of proteins were calculated by densitometry and listed beneath the bands. Results are presented as means ± SEM. ** *p* < 0.01; *** *p* < 0.005 vs. MCF7; ## *p* < 0.01 vs. MDA-MB-231. The original western blots can be found in Appendix A.

**Figure 5 cancers-14-01487-f005:**
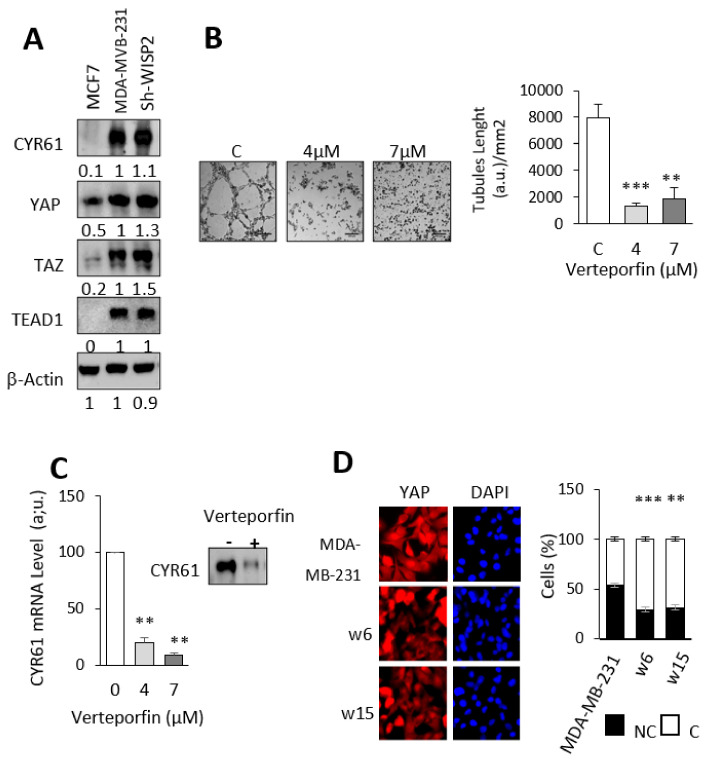
WISP2 regulates the Hippo/Yap pathway. (**A**) Representative Western blot analysis of Hippo/Yap markers in the different breast cancer cell lines. β-Actin was used as loading control; Levels of proteins were calculated by densitometry and listed beneath the bands. (**B**) MDA-MB-231 cells were treated with 4 and 7 µM of Verteporfin for 24 h and subjected to tube formation assay for 24 h. Length of tubes was measured; Scale bar = 200 μm. (**C**) CYR61 mRNA expression was detected by qRT-PCR and secreted protein by Western blotting in the MDA-MB-231 cell line treated with verteporfin; (**D**) Representative images of YAP immunostaining in the MDA-231-cell line and in MDA-MB-231 cells stably transfected with WISP2 (clone w6 and w15). YAP subcellular localization was quantified and reported as labeling observed in nucleus and cytoplasm (NC; Black) or in cytoplasm alone (C; White). Results were obtained in three separate experiments in which 4 images corresponding to 200 cells approximately were examined. All data are represented as ± SEM. ** *p* < 0.01; *** *p* < 0.005. The original western blots can be found in Appendix A.

**Figure 6 cancers-14-01487-f006:**
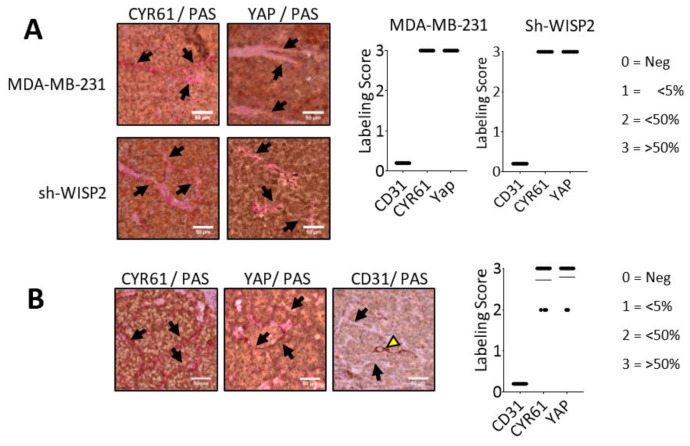
CYR61, YAP, and CD31 expression in mouse xenografts and in triple negative human breast cancer. (**A**) Thick paraffin sections of breast obtained from tumor xenografts of mice injected with MDA-MB-231 or sh-WISP2 human breast cell lines were subjected to CYR61/PAS and YAP/PAS double staining (magnification 40×). PAS staining revealed VM channels which are lined by tumor cells positive for Periodic Acid-Schiff and CYR61 or YAP (Black Arrow). (**B**) Thick paraffin sections obtained from triple negative human breast tumors were subjected to CYR61/PAS, YAP/PAS and CD31/PAS double staining (magnification 40×). Representative images of IHC staining show strong expression of CYR61 and YAP proteins in cells that form vascular mimicry tubules revealed by PAS staining, whereas CD31 is not detected in these structures (black arrow). CD31-positive endothelial vessel (yellow arrow) indicates regular vascular channels positive for CD31 staining. A protein expression score was expressed according to the intensity of the observed labeling. Scale bar = 50 μm.

## Data Availability

The data presented in this study are available in this article (and Appendix A).

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
