# Peer review of "WISP2/CCN5 Suppresses Vasculogenic Mimicry through Inhibition of YAP/TAZ Signaling in Breast Cancer Cells"

_cancers, 2022, doi:10.3390/cancers14061487_

Round 1
Reviewer 1 Report
The findings are interesting and technically well performed. Specific points that the authors need to address are as follows:
- Most of the key experiments have been done in MDA-MB-231 cell line. Addition triple negative breast cancer cell lines should be used to validate the key findings of the study.
- It is not clear that how the loss of WISP2 expression can lead to a significant increase of CYR61 expression.
- The authors should provide their own justification and relevance of the study. This will help the readers to understand the importance of the paper.
- The conclusion section should be improved and therapeutic options that can be designed to suppress VM formation in breast cancer should be highlighted.
- All sections of the manuscript should be checked in terms of typographical errors.
Reviewer 2 Report
- Authors have not mentioned the statistical details in Figure 3 caption. No information about *, ***, ***.
- Size and resolution of image is poor. Need to replaced with high quality image.
- Introduction. Author must support the rational and objective of the work design.
- Overall the manuscript needs careful revision of error in sentence/superscript/subscript.
- Rewrite the conclusion with important findings of the study.
